# The Impact of Obesity on Inflammatory Bowel Disease

**DOI:** 10.3390/biomedicines11123256

**Published:** 2023-12-08

**Authors:** Patricia Kaazan, Warren Seow, Shaanan Yong, Leonie K. Heilbronn, Jonathan P. Segal

**Affiliations:** 1Faculty of Health and Medical Sciences, School of Medicine, University of Adelaide, Adelaide, SA 5005, Australiashaananting.yong@student.adelaide.edu.au (S.Y.);; 2South Australian Health and Medical Research Institute, Adelaide, SA 5000, Australia; 3IBDSA, Tennyson Centre, Kurralta Park, SA 5037, Australia; 4Department of Surgery, Flinders Medical Centre, Adelaide, SA 5042, Australia; 5Department of Gastroenterology, Royal Melbourne Hospital, Melbourne, VIC 3052, Australia; segaljonathan0@gmail.com; 6Department of Medicine, University of Melbourne, Melbourne, VIC 3052, Australia

**Keywords:** IBD, obesity, mesenteric adipose tissue, creeping fat

## Abstract

Obesity is prevalent in the inflammatory bowel disease (IBD) population, particularly in newly developed countries where both IBD and obesity in the general population are on the rise. The role of obesity in the pathogenesis of IBD was entertained but results from available studies are conflicting. It does, however, appear to negatively influence disease course whilst impacting on our medical and surgical therapies. The pro-inflammatory profile of the visceral adipose tissue might play a role in the pathogenesis and course of Crohn’s Disease (CD). Interestingly, isolating the mesentery from the surgical anastomosis using a KONO-S technique significantly decreases anastomotic recurrence rate. Anti-obesity therapy is not widely used in IBD but was suggested as an adjunctive therapy in those patients. In this review, we aimed to highlight the epidemiology of obesity in IBD and to describe its influence on disease course and outcomes.

## 1. Background

Obesity has become one of the main threats to global health, gaining significant clinical and scientific attention and causing a significant economic burden. Most countries have witnessed escalation in obesity prevalence over the last three decades [1,2].

According to the World Health Organisation (2016), 13% of adults live with obesity and 39% are overweight; a proportion that has nearly tripled since 1975 [2]. According to the Australian Institute of Health and Welfare report in 2018, 31% of Australians live with obesity; a proportion that has almost doubled since 1995 (19%) [3].

The food environment and the associated alterations in the normative eating behaviours alongside a reduced need for daily physical activity are the major drivers of these changes [4].

Over 40% of the Australian diet is derived from ultra-processed foods [5]. The food industry is marketing convenient, highly processed food, from cheap agricultural inputs, that result in textural changes to food and affect the way food is digested and absorbed [6].

Obesity is well reported as a chronic inflammatory state. Hypertrophic adipose tissue is infiltrated by multiple immune cell subtypes which produce a pro-inflammatory profile and produces large amounts of pro-inflammatory gene mediators [7,8,9]. Inflammatory bowel disease (IBD) is a group of chronically relapsing, immune-mediated conditions of the gastrointestinal tract such as Crohn’s disease (CD) and Ulcerative colitis (UC), that also involve genetic and environmental factors [10].

The epidemiology of IBD is rapidly changing and a shift in the epidemiological pattern has been reported [11]. The incidence of IBD is increasing or stable in virtually every region of the world. It is most common in the Western world, affecting 0.5% of the general population; similarly, the prevalence is highest there ranging from 10 to 30 per 100,000 [11].

An increase in the age-standardized prevalence rate of IBD in regions that formerly had low prevalence, such as East and South Asia, Oceania, and sub-Saharan Africa has been reported [12].

It is likely due to a combination of factors such as improvements in the socioeconomic status of newly industrialized countries, changes in diet and lifestyle factors, improved sanitation, changed microbiota, and environmental factors.

The obesity rate in IBD parallels that of the general population [13]. There is some data to suggest the deleterious effects of obesity on the pathogenesis, the natural history, and the response to therapy in IBD. While obesity is characterized by a widespread increase in adipose-tissue hypertrophy and a chronic low-grade inflammation, the fat accumulation in CD, specifically, is more localized, has different metabolic characteristics, and is independent of the total body weight. This visceral adipose tissue (VAT) has been implicated in the pathophysiology of CD [14,15,16].

In this review, we discuss the epidemiology and pathophysiology of obesity in IBD. We also discuss the effects of obesity and visceral adipose tissue on disease course, treatment response, and surgical outcomes, and whether anti-obesity therapy is safe and beneficial in patients with IBD and obesity.

## 2. Prevalence of Obesity in IBD

The worldwide burden of IBD has increased markedly over the last few years, at least partly due to the compound prevalence of this disease [17].

Adult studies in patients with IBD show that around 15 to 40% are obese and another 20 to 40% are overweight [18,19].

Unpublished data from the Australian cloud-based IBD-specific medical record CrohnsColitisCare (CCCare) suggest that 22% of patients with IBD are obese and 50% are overweight based on their body mass index (BMI).

A similar trend was observed in paediatric patients with IBD. In a multiple-site registry of 1598 children with IBD, Long and colleagues observed that 23.6% of children were above the 85th sex-specific BMI-for-age percentile including 9.5% of children who were above the 95th percentile [20].

These epidemiological data suggest that the rate of obesity in patients with IBD is similar to that of the general population with limited information regarding a causality or directionality effect between the two chronic diseases.

## 3. Obesity and IBD Pathogenesis 

The role of obesity as a cause of IBD remains understudied (Table 1).

Premorbid obesity has been linked with an increased risk of developing Crohn’s disease, although this observation is not consistent across studies.

In the Nurses’ health study, BMI at age 18 was predictive of risk of developing CD (BMI > 30 versus normal BMI: HR 2.33, 95% CI 1.15–4.69) but not UC (HR 1.17, 95% CI 0.54–2.52) [21].

The effect of obesity on risk of IBD development might be age dependent, with obesity in teenage and young adulthood associated with a higher risk of developing CD. In a population-based study registry, patients hospitalized with a primary diagnosis of obesity, the relative incidence of Crohn’s disease is highest when obesity is diagnosed before 30 years of age (standardized incidence rate (SIR) 1.92, 95% CI 1.17–2.96), and decreases with increasing age at obesity diagnosis [28].

The Copenhagen School Health Records Register suggests that childhood obesity, as defined by a BMI > 31.0 kg/m^2^, increases the risk of CD but not of UC [22]. On the contrary, The European Prospective investigation into Cancer and Nutrition study found no association between BMI and incident CD or UC [23].

The UK Biobank followed 492,998 patients without IBD over a period of 12.5 years and found a positive association between adiposity, as measured by BMI and waist circumference, and the risk of incident IBD; reporting that these associations could be partially mediated by unhealthy metabolism and active inflammation [24].

These epidemiological studies have significant limitations, for example, they are cross sectional, which limits causality assessment. BMI is an imperfect screening tool as it does not provide information on fat versus muscle mass. It is also important to note the heterogeneity in relative body shape and composition across race and ethnic groups, genders, and age categories. A lower BMI defines obesity for South East Asians [29], Indians and Aboriginal people [30].

BMI as a sole surrogate does not capture obesity’s association with IBD given the established immunological activity of “the visceral adipose tissue”.

Fat mass and distribution play an important role in obesity-related cardiometabolic health outcomes, and waist circumference appears to be a better surrogate than BMI for risk of cardiometabolic disease and for IBD [31,32,33].

Magnetic resonance imaging and Dual-energy X-ray absorptiometry (DEXA) are more precise direct measurements of adipose tissue volume and distribution, hence they should be considered in future studies [34].

### 3.1. The Adipose Tissue

For a long time, the adipose tissue (AT) has been regarded as an energy store with some endocrine functionality. However, we now understand that it has additional important immunological, metabolic, and regulatory functions. The subcutaneous and visceral adipose tissue compartments display distinct metabolic and immunological profiles [35].

Placental mammals have three main types of adipocytes: white, beige, and brown, organized into discrete deposits throughout the body. White adipocytes are specialized in lipid storage and release. Beige and brown adipocytes are thermogenic cells able to expend nutritional energy into heat. The white adipose tissue (WAT) has a “metabolic plasticity” controlled by hormonal and neuronal signals allowing it to switch between two opposite metabolic programs [36].

WAT is an essential endocrine organ with an unparalleled capacity to expand. An expansion in response to caloric stimuli is an appropriate physiological adaptation; however “obesity“ is associated with an increased risk of cardiometabolic disease [37]. Individuals who preferentially expand intra-abdominal WAT are at a greater risk of metabolic syndrome then those who accumulate subcutaneous WAT. The detrimental effects of abdominal WAT can be at least partially explained by its proximity to portal circulation. Healthy obesity is also associated with adipocyte hyperplasia (smaller size and higher number).

Obesity is a state chronic low-level inflammation, and the AT is the main source of immune-mediated inflammation in obesity. The normal AT relies on a considerable amount of oxygen from surrounding vasculature; hypertrophied adipocytes in obesity induce hypoxic conditions by compromising surrounding blood vessels. Chronic hypoxia can ultimately lead to cell death by amplifying inflammatory signals and regulating recruitment and differentiation in the immune cell population. For example, macrophages infiltrate to scavenge dead adipocytes in the involved fat deposit, amplifying a local inflammatory response by producing TNF a and IL6 [38,39].

The visceral adipose tissue (VAT) or white mesenteric adipose tissue are metabolically active organs, housing multiple immune cells that are the source of inflammatory components such as TNF-a and IL1 which are at the centre of the development of IBD [15].

### 3.2. Creeping Fat…Good and Bad 

Creeping fat (CF) is a locally restrictive form of VAT (Figure 1) where mesenteric fat hyperplasia (rather than hypertrophy) is limited to areas of inflamed and fibrotic bowel in CD, covering more than 50% of the bowel circumference and leading to the loss of the bowel–mesentery angle [16,40]. It is thought to be more immunologically active than other VAT, and its extent correlates with the extent of transmural inflammation, fibrosis, muscle hypertrophy, and stricture formation [41,42]. CF is not seen in UC.

It appears that there is a relationship between bacterial translocation and the development of CF [14]; when the gut microbiota creep into fat, the fat creeps back. Humoral and cellular alterations within the CF are unique and differ from those observed in hypertrophied fat tissues in obesity [43].

The previously under-appreciated diversity of microbes in the CF surrounding the GI tract was described by Ha et al. [44]. The CF harbours a bacterial consortium defined by “ Clostridium innocuum”. Single-cell RNA sequencing characterized CF as both a pro-inflammatory and pro-adipogenic milieu, the expansion of which prevents systemic bacterial translocation. CF itself and the underlying intestinal tissue tend to be fibrotic. Whether CF fibrosis drives intestinal fibrosis remains unknown.

The adipose tissue is almost always consistently histologically encroaching into the underlying ileal muscularis. The axial polarity and the predominance of the ulcerations beneath the attachment of the mesentery are characteristics of CD but not of intestinal tuberculosis and Crohn’s-mimicking infective diseases [45].

An important study by Mao et al. [46] concluded that activated muscularis propria cells produce Fibronectin which in turn stimulates pre-adipocytes migration from the mesenteric tissue and its differentiation into adipocytes forming the CF.

### 3.3. Adipocytes

Adipocytes’ emergence from obscurity began in 1987 [47].

They function like macrophages, surveying their environment for microbial products, mediating innate immune responses while pre-adipocytes have intrinsic phagocytic properties [48]. They have a certain degree of plasticity.

The absolute number of adipocytes is genetically determined [40], and although reasonably stable in adulthood, some people might remain hyperplastic expanders while others are hypertrophic. CF is the result of adipose-tissue hyperplasia rather than hypertrophy [40].

They release several bioactive polypeptides called “adipokines” (such as adiponectin, Leptin and resistin) that are able to modulate “body weight”, appetite, glucose homeostasis, inflammation, and blood pressure. They are also able to produce pro-inflammatory cytokines (such as TNF-a, IL6 and IL1), particularly in the chronic sub-inflammatory state of obesity as well as anti- inflammatory cytokines [49].

TNF-a is a crucial disease mediator involved in the pathophysiology of IBD, and therapy with anti TNF-α monoclonal antibodies has been shown to induce remission in active CD [43].

Interestingly, the levels of adiponectin and leptin were shown to be increased in the mesenteric fat of patients with Crohn’s. Levels of resistin correlate with Crohn’s disease severity and disease activity, further supporting the interaction theory between adipokines and adipocytokines to produce mucosal inflammation [16].

## 4. Obesity and Disease Course 

Although the data are sparse and conflicting, it appears that obesity might be associated with poorer disease outcomes, remission rates, and quality of life, and increased unplanned healthcare utilization and risk of surgery.

Results from the Swiss inflammatory bowel disease cohort show that obesity is associated with decreased rates of disease remission and an increased risk of complicated disease course in patients with CD but not UC over a six-year follow-up period [50].

In a nationwide readmissions database study from the United States, obesity in IBD was associated with an increased early readmission, characterised by early system and patients’ burden, despite the availability of weight-based therapies [51].

A retrospective study of 148 patients with CD has shown that those with a BMI > 25 kg/m^2^ were older at diagnosis had an earlier time to first surgery [27].

Another retrospective French cohort study suggested more frequent perianal complications and marked year by year disease activity without alterations in the disease course in patients who were obese. Interestingly the rate of obesity in patients with IBD at the time of this study in 2002 was 3% [52].

These results are inconsistent and vary across studies.

Data from a prospective registry of Crohn’s patients showed a lower prevalence of penetrating disease activity among individuals with obesity and Crohn’s disease, but a comparable rate of perianal disease or surgery vs. non-obese Crohn’s patients with obesity [26].

In another cross-sectional study of patients with CD, there was no difference in the prevalence of penetrating or stricturing complications between individuals who were obese and those of a normal weight [18]. Obesity has not been associated with disease severity in patients with UC; in contrast, retrospective data suggest that overweight patients might have a less complicated disease course [21,25]. Of note, a major limitation of the aforementioned studies is the use of the BMI as a surrogate marker of adiposity.

## 5. Obesity and Disease Management 

It seems logical to assume that obesity impacts the medical management of IBD as it affects the absorption, distribution, and clearance of the current IBD drugs. Such an effect was demonstrated by few pharmacokinetic studies [53,54].

The mode of drug administration (subcutaneous vs. intravenous) along with the issue of weight-based vs. fixed dose regimens seem to be leading factors through which obesity impacted upon response to medical therapy. Differences have been described in the pharmacokinetic profiles of subcutaneous vs. intravenous therapy [55], leading to the suggestion that obesity could potentially influence subcutaneous therapies more than intravenous ones.

## 6. Obesity and Thiopurines

Thiopurines have been used to maintain remission in IBD for decades. Adipose tissue may inhibit the response of azathioprine (AZA) [56].

A retrospective study showed a differential response to AZA where UC patients with a BMI of >25 had poorer responses to AZA as compared to patients with a BMI of <25 but no correlation was shown between BMI and response to AZA in patients with CD [56]. Poon et al. reported that there was an association between patients with a BMI > 25 and sub-therapeutic 6-thioguanine levels [57], and that patients with a BMI > 30 had a higher 6-MMP:6-TGN ratio resulting in suboptimal thiopurine metabolism.

## 7. Obesity and Anti-TNF Therapy

Tumour necrosis alfa blockers remain important agents in the induction of remission in both CD and UC and are the cornerstone of therapy in fistulizing and perianal CD.

Infliximab (IFX) is a weight-based intravenous TNF- α antagonist and Adalimumab (ADA) is a subcutaneous non-weight-based TNF- α antagonist. Weight, especially adiposity, affects the pharmacokinetic properties of many drugs, such as drug clearance, volume of distribution, and elimination half-life.

In a retrospective biologically naïve IBD cohort treated with IFX [58], obesity seems to be associated with an earlier loss of response to IFX therapy; equally, it may decrease the response to therapy overtime in ADA-treated patients [59]. In a single-centre retrospective cohort study, Bhalme et al. observed a higher likelihood of ADA, but not IFX, dose escalation in patients who had a BMI of at least 30 kg/m^2^ than in patients who were not obese [60]. A possible explanation is the fixed dosing schedule of ADA, resulting in obese patients being less likely to receive weight-appropriate therapy.

A large retrospective analysis of 1494 IBD patients compared the actual milligram per kilogram dose of medication that patients were receiving based on their BMI category. As BMI increased, the actual weight-based dose that patients received significantly decreased for both S/C- and IV-delivered anti-TNF therapies. For instance, the mean weight-based doses of infliximab and adalimumab were 3.96 mg/kg and 0.31 mg/kg, respectively, in patients with class III obesity, compared to 7.89 mg/kg and 0.62 mg/kg, respectively, in normal-weight patients (*p* < 0.0001) [19]. A plausible explanation is the lack of dose adjustment in accordance with increasing weight due to the absence of symptoms in patients who are in remission, although this is contradicts prior studies suggesting an earlier loss of response to anti-TNF therapy in obese patients.

The REMSWITCH study showed that switching from IV to SC IFX is safe and well accepted, leading to a low risk of relapse except for those receiving 10 mg/kg every 4 weeks requiring double the dose fortnightly [61].

Patients with obesity and IBD are prone to a TNF-sink phenomenon where TNF- α blockers are sequestered within the adipose tissue-secreted TNF-a; high body weight appears to be a risk factor for increased drug clearance, resulting in short half-life and low trough drug concentrations as shown in population pharmacokinetics studies involving IBD-approved biological therapies such anti-TNFa and anti-integrins [58].

## 8. Obesity in IBD-Related Surgery

Traditionally, obesity was considered a relative contraindication for surgery, owing to the technical complexity and the higher rates of complication such as wound infection, anastomotic leakage, and stoma failure [62]. However, with the growing demand of patients with obesity being referred for IBD surgery, the technical expertise for such procedures in this population has improved over the years.

Restorative proctocolectomy (RPC) and ileo-pouch anal anastomosis reservoir (IPAA) are the gold-standard bowel reconstruction methods in patients with ulcerative colitis requiring surgery. The reconstruction is performed through an anastomosis between the anal canal and the ileal pouch, commonly constructed as a J-pouch in the distal ileum. One of the technical challenges relating to obesity is the formation of an ileocolic pouch reservoir. Attaining the sufficient length of bowel required to construct a tension-free ileal pouch in the pelvic floor may be challenging due to the thickened and shortened mesentery layer in a patient with obesity. In a retrospective study of 130 patients undergoing RPC and IPAA for UC, Tzatzarakis and colleagues found significantly higher rates of conversion to open surgery in the patient cohorts with obesity, owing predominantly to a short mesentery (*p* = 0.006) [63]. Notably, the majority of patients with obesity and a short mesentery underwent a three-stage procedure, suggesting that previous operations may have contributed to more abdominal adhesions and a contracted mesentery as well. To mitigate this, common intraoperative practices involve a transverse incision in the peritoneal layers of the small bowel mesentery and ligating the ileocolic artery to widen the reach of small bowel when constructing an the ileoanal pouch reservoir. Kiran and Fazio et al. also adopted other manoeuvrers in patients with obesity when surgeons struggle to gain sufficient mesenteric length for the ileal pouch [64]. These manoeuvres include the release of small bowel mesentery from the retroperitoneum, a higher ligation of ileocolic vessels, mobilisation of the duodenum, incisions along mesenteric edge of the small bowel to release peritoneum, and excision of any redundant mesenteric tissue lateral to the superior mesenteric vessels. Ligation of the main trunk or branches of superior mesenteric artery have also been discussed in the current literature; however, such a manoeuvre confers a higher risk of whole bowel ischaemia.

Creation of an “S” instead of a “J” pouch may also be attempted when there is inadequate length of small bowel mesentery to reach the top of the anus, allowing for more length in the “S” out-spout of distal small bowel attached to the anal canal. However, in circumstances when the ileum is not able to reach the pelvic floor despite the aforementioned manoeuvres, it may be necessary to perform a colectomy with end ileostomy [65]. Over time, the mesentery of the ileum commonly elongates, subsequently allowing for sufficient reach.

Low anastomosis in obese patients with narrow conical pelvises may also prove challenging. Preoperative strategies, such as weight loss, will provide better access to perform deep pelvic anastomosis. Weight loss counselling for morbidly obese patients prior to bowel reconstruction has advantages; however, there is no evidence in the current literature highlighting the beneficial outcomes of preoperative weight loss before surgery [66].

Another technical challenge relating to obesity is the construction of a loop ileostomy. A loop or diverting ileostomy is commonly performed to reduce the risk of anastomotic leak or pelvic sepsis in most of the patient population with obesity. However, in patients with severe obesity and large abdominal walls, the ileostomy may be under undue tension which may subsequently retract the stoma. If the standard location of stoma formation is not feasible, an alternative loop stoma through the inferior edge of the midline incision may be utilised for patients with obesity undergoing RPC and IPAA. It is well established, however, that this confers a higher risk of wound infection and a suboptimal visualisation of the stoma for patients to manage at home [65]. Other strategies described by Meguid et al. suggest constructing a wider tunnel for the stoma opening by removing subcutaneous adipose tissue via a lipectomy at the planned ileostomy site [67]. A subcutaneous tissue flap can be constructed as an alternative approach to facilitate exteriorising the ileostomy in two steps: through the abdominal fascia, followed by bringing the bowel out through an aligned aperture in the subcutaneous tissue Duchesne [68]. Laparostomy devices, including a self-expanding wound protector (Alexis^®^ wound protector, Applied Medical, Rancho Santa Margarita, CA, USA), have also been used as a manoeuvre to create a conduit for exteriorising the bowel for ileostomy.

Recent evidence indicates that creeping fat is an independent risk factor for postoperative disease recurrence in patients with CD [69]. Notably, the association of CF with expanding mesenteric adipose tissue and inflammatory markers, transmural inflammation, and a thickened intestinal wall may corelate with a more complex disease process in CD [70,71]. Instinctively, mesentery-based surgeries may improve disease outcomes by circumventing these pro-inflammatory interactions. The inclusion of the mesentery in ileocolic resection, as described by Coffey et al., showed a significant reduction in surgical recurrence rate in patients with CD [72]. Specifically, 40% of the 30 patients who underwent mesentery-sparing resection required CD-related reoperation; whereas 2.9% of the 34 patients who underwent mesentery-included resection required reoperation. On the contrary, a recent meta-analysis investigating the efficacy of Kono-S, a surgical approach involving anti-mesenteric, functional, continuous, hand-sewn anastomosis with preservation of the mesentery, yielded similar reduced surgical recurrence rate [73]. Notably, the postoperative recurrence rate in patients operated on with the Kono-S technique was 3.4% vs. 15% in the standard anastomosis group. Despite contradicting evidence suggesting both mesentery-sparing Kono-S and radical mesenteric resection confers improved long-term outcomes, an argument can be made that both techniques isolate the anastomosis as far away as possible from the affected mesenteric tissue which may explain this paradox [73].

The current literature highlights the poorer short- and long-term outcomes in patients with obesity undergoing bowel reconstruction surgery for IBD. In a nation-wide US study of over 380,000 patients with IBD, Jain et al. found that patients with obesity and concomitant IBD had statistically significant increased rates of overall postoperative complications, including wound infection (OR 1.35, *p* = 0.01), pulmonary complications (OR 1.16, *p* = 0.02), and intra-abdominal sepsis (OR 1.30, *p* = 0.02) [74]. On the contrary, there were no differences in venous thrombotic events, cardiovascular complications, and death. Stoma complications in patients with severe and complex obesity are also not uncommon, and patients with concomitant IBD are reported to have higher rates of stoma complications, predominantly stoma herniation, high output volume, and bowel ischemia [68,75]. Similar findings were reflected by Benoist et al., who found that patients with obesity had higher rates of anastomotic leak and post-operative haemorrhage [76]. Thus, despite the challenges of creating a diverting stoma in patients with obesity, the risk of postoperative complications precedes the necessity to proceed with a two-stage operation.

The increased risk of postoperative infection is a crucial complication to consider in patients with obesity, owing to the frequent use of immunosuppressive agents, particularly TNF- α inhibitors [77]. In a retrospective study comparing 301 chronic ulcerative colitis patients undergoing ileal pouch anal anastomosis (IPAA), who were preoperatively treated with TNF- α inhibitors, increased postoperative wound infection rates amongst patients with higher BMIs treated with TNF- α inhibitors within 2 months of IPAA were observed [78]. Beyond isolated therapies, the rates of infection have also been shown to increase in combinative therapy in patients with or without obesity, particularly those administered higher supplementary doses of prednisolone.

Interestingly, more recent studies have showed that visceral adiposity, rather than overall high BMI, may be independent predictors of poor surgical outcomes in obese patients undergoing IBD surgeries. In a retrospective cohort study of 164 patients with active CD undergoing surgery, Ding et al. found an independent 2.7-fold increase in overall postoperative complications in patients with high visceral fat area of over 130 cm^2^ [79]. Moreover, another study utilizing analytic morphomics to assess body fat composition in patients with CD highlights the increased wound infection rates in high visceral fat area rather than BMI alone [80].

## 9. Obesity Treatments and Outcomes in IBD

### 9.1. GLP1-Based Therapy

Incretins such as glucose-dependent insulinotropic polypeptide (GIP) and glucagon-like peptide-1 (GLP-1) are hormones whose main physiological role is augmenting insulin secretion and delaying gastric emptying after their nutrient-induced secretion from the gut [81].

Glucagon-like peptide 1 receptor (GLP-1R) mimetics such as Exenatide, liraglutide and Semaglutide are all approved, effective anti-obesity medications [82].

Despite an increase in the prevalence of obesity in IBD over the last decade, the utilization of anti-obesity pharmacotherapy has remained limited, possibly because of the GI side effects reported. In a retrospective analysis of population-level data using a commercial database from 2010 to 2019, only 2.8% of eligible adults with IBD and obesity were prescribed anti-obesity pharmacotherapy in the last 10 years, with trends increasing from 1.4% in 2010 to 3.6% in 2019 [83].

GLP1 based therapies are well established anti-obesity pharmacotherapy but the disease course of IBD following these therapies remains unclear. In a nationwide Danish registries study, 3751 patients with IBD and T2DM treated with antidiabetic agents were identified between 2007 and 2019. There was a lower risk of adverse clinical events (such as the need for steroids, biological therapies, and hospitalisations) amongst patients treated with GLP1-based therapies compared with other antidiabetics. These findings suggest that GLP-1 based therapies might improve disease course in IBD [84].

Recently, a dual GIP/GLP-1 receptor co-agonist (Tirzepatide) has been approved for Type 2 Diabetes [85]. Beyond its glycaemic effects, it has a profound effect on body weight [86] and waist-circumference reduction, and it appears to have less side effects than GLP1-RA. GIP has a direct action on the adipose tissue and its resident immune cells.

### 9.2. Bariatric Surgery

Bariatric surgery is reported to be the most effective option for weight management in patients with severe, complex obesity with the trend of weight loss surgery dramatically increasing in the past decade. There are two broad categories of bariatric surgery—restrictive and malabsorptive procedures. Restrictive procedures, such as gastric banding and sleeve gastrectomy, were developed to promote weight loss by having the patient experience early satiety during food intake by partitioning the stomach, thereby creating a small chamber to store the food bolus consumed. Gastric banding involves the insertion of an adjustable and inflatable band around the proximal stomach which is serially filled to achieve restriction via a subcutaneous access port. Sleeve gastrectomy (SG) involves the transection and removal of the greater curvature portion of the stomach, leaving the remaining narrowed stomach along the lesser curvature. Although SG is considered as a restrictive procedure, it is hypothesised to induce hormonal changes such as a reduction in serum ghrelin levels, which helps promote early satiety and prolonged satiation. Current practices of Roux-en-Y gastric bypass (RYGB) consist of a small gastric pouch of 15–30 mL being anastomosed to the distal jejunum, bypassing the bilio-pancreatic secretions. Beyond its combined restrictive and malabsorptive mechanisms, RYGB is now recognised to have additional benefits, including nutrient restriction and hormonal and neuronal manipulation. Owing to these multi-modal mechanisms, RYGB has established itself over the last three decades as a durable operation with excellent weight loss and metabolic syndrome resolution.

Bariatric surgery confers higher rates of short and long-term complications but provides remission from type 2 diabetes and weight loss [87]. There is sparse evidence on the outcomes of bariatric surgery in obese patients with IBD and the potential concomitant risk and complications—mostly derived from a small cohort of case reports [62].

As previously discussed in this paper, inflammatory markers associated with obesity are observed to be raised more in Crohn’s disease. As such, we can hypothesise the suitability of bariatric surgery for refractory IBD symptoms, or following the failure of other medical therapies [27]. Another case report by Lascano et al. presents a 39-year-old, morbidly obese male with a longstanding history of UC and hypertension. The patient’s UC was controlled by oral and rectal mesalamine and azathioprine, in addition to varying doses of prednisolone for worsening acute UC flare-ups. After a laparoscopic Roux-en-Y gastric bypass, he achieved significant relief from his UC-related diarrhoea, tenesmus, and pyoderma gangrenosum alongside his successful weight loss. Since his bypass, his UC was controlled without mesalamine and with the tapering of his prednisolone and azathioprine doses [88]. Consistently, a more recent case series by Keidar et al. reports a 71% reduction in IBD pharmacotherapy following 10 IBD patients who underwent bariatric surgery, and an over 40% cessation rate in the maintenance medical IBD therapy in the 2-year follow-up period [89]. Overall, recent studies have suggested that the reduction in low-grade systemic inflammatory mediators as a result of bariatric surgery and weight loss can be attributed to the success in reducing the active IBD.

Despite the proposed success of bariatric surgery, there may be negative implications in altering small bowel anatomy in IBD patients. Several case series describe the consequential malabsorptive state in patients with IBD, suggesting a relationship between the segments of resected small bowel mucosa and the formation of new or worsening Crohn’s disease. Tenorio Jimenez et al. reported a 39-year-old morbidly obese male with a history of metabolic disease, thrombophilia, and UC who subsequently developed protein malnutrition 10 months after a biliary–pancreatic diversion requiring repeated long-stay hospitalisation [90]. Moun et al. presented a case of a 40-year-old morbidly obese female with medically managed ileocolic CD and associated type 2 diabetes and hypertension. She underwent a Roux-en-Y gastric bypass with successful weight reduction but developed active recurrent Crohn’s disease 8-weeks post-operatively [91]. The authors proposed that the worsening immune response observed in this case was due to the inappropriate interaction between the altered gut microbiome and intestinal epithelial host receptors. It is well established that bacterial overgrowth may develop as a consequence of gastric bypass surgery, and this complication could be exacerbated in IBD patients genetically predisposed to intestinal inflammation.

Common complications following laparoscopic Roux-en-Y gastric bypasses includes anastomotic strictures, ileus, and the formation of fistulas and abscesses. Such complications are prevalent in both disconnected diseased segments of intestines and the site of the reduction plasty in the stomach. Due to the nature of bariatric surgery, i.e., Roux-en-Y bypass, which involves surgical resection of small intestinal mucosa, the aforementioned complications are most profound in the diseased small bowel segment operated [92]. To date, there is a lack of evidence comparing surgical outcomes in bariatric surgery between small bowel versus large bowel predominant Crohn’s disease.

Another aspect to consider during post-bariatric surgery in IBD patients is the challenge to performing subsequent surgeries for intestinal disease, such as a colectomy with reservoir formation. Undoubtedly, the disruption of the small bowel and mesentery anatomy after a gastric bypass may affect technical issues of construction, accessibility, and function of an ileoanal pouch reservoir in UC patients [93]. Additionally, the postoperative risk of adhesions and scarring will likely decrease the chances of successful outcomes in subsequent operations.

Nonetheless, tailoring the appropriate bariatric procedure involves delicate patient selection through a multidisciplinary team approach [89]. Current data supports that the least invasive bariatric surgery should be offered, and gastric reduction plasty should be the preferred option for morbidly obese patients [94].

## 10. Physical Exercise as a Treatment for Patients with IBD

Physical activity was shown to improve health-related quality of life (HRQOL) in patients with chronic diseases and IBD [1].

Bilksi et al. demonstrated that moderate physical activity does not have any negative side effects on stable IBD patients. Physical activity improves bone mineral density, a common complication in patients with IBD, and reduces the risk of osteoporotic fractures [95]. Further research is still required to determine the amount and intensity of physical exercise IBD patients need to optimise the benefits. Different types of exercises also need to be evaluated to identify their effectiveness in treating patients with IBD.

## 11. Conclusions

The prevalence of both obesity and IBD are on the rise worldwide, particularly in newly industrialized countries. Although a pathobiological linkage between the two conditions is plausible through common environmental risk factors such as high fat low fibre diet, intestinal microbiome alterations and a metabolically active visceral adipose tissue, epidemiological studies associating obesity with the development of IBD remain relatively limited.

The effect of obesity on disease course and activity, and on response to therapy, is believed to be negative; however studies are contradictory and heavily reliant on BMI as a surrogate marker of adiposity. The use of anti-obesity therapy as indicated in IBD patients remains limited; although, reassuringly, GLP-1 based therapies appear to be safe and beneficial in obese patients with IBD and this should be encouraged in clinical practice.

Longitudinal prospective studies are difficult to perform as they require many patients and long periods of follow up; however simple measures such as implementing routine measurement of waist circumference in day-to-day IBD practice might improve the quality of cross-sectional studies.

Future studies are required to advance our understanding of the complex interactions between IBD and obesity.

## Figures and Tables

**Figure 1 biomedicines-11-03256-f001:**
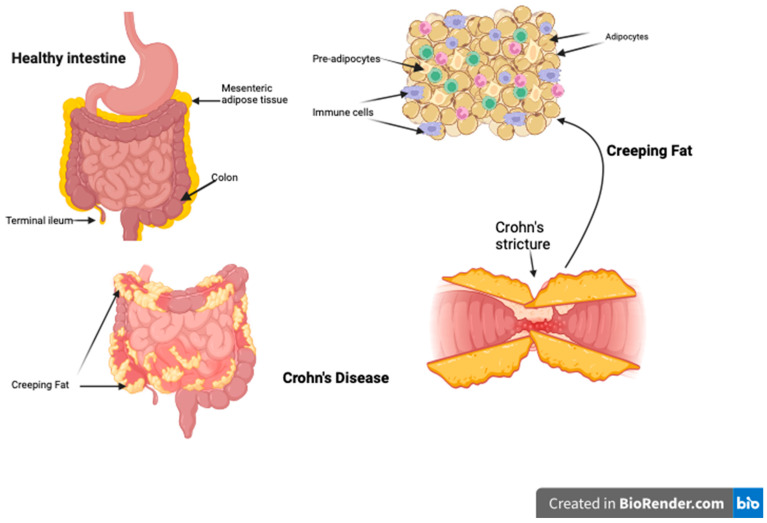
Creeping fat surrounding the inflamed intestine. It is composed of hyperplastic adipocytes and pre-adipocytes and a multitude of immune cells. Those cells produce inflammatory mediators which contribute to fat function/malfunction, hence modulating intestinal inflammation.

**Table 1 biomedicines-11-03256-t001:** Key studies on epidemiology of obesity (BMI ≥ 30 kg/m^2^) and the associated risk of IBD.

Study	Type of Study	Country	Year	Patient Characteristics (n; % Females)	Prevelance of Obesity	Median Follow-Up (Years)	Key Findings
Khalili et al. (2014) [21]	Population-based cohort study	USA	1989	CD: 153; 100% UC: 229; 100%	CD: 5.9% obese UC: 3% obese	18	Increased weight gain at age 18 was associated with increased risk of CD (weight gain > 13.6 kg vs. <2.3 kg: HR 1.52, 95% CL 0.87–2.65) Anthropometric measures of adiposity were associated with an increased risk of CD but not UC (obese vs. normal BMI at age 18: HR 1.17, 95% CL 0.54–2.52) Approximately 2-fold increased risk of developing CD in women with obesity compared to women with normal BMI age 18 (HR 2.23, 95% CL 1.15–4.69) post-adjustment for confounding variables such as physical activity
Jensen et al. (2018) [22]	Population-based cohort study	Denmark	1977	CD: 1500; 58% UC: 2732; 54% female	CD: 5% obese at age 13 UC: 5% obese at age 13	39	Direct association between childhood BMI and CD diagnosed before 30 years of age Inverse association between childhood BMI and UC irrespective of age
Chan et al. (2013) [23]	Prospective cohort study	Denmark Germany Netherlands Italy Sweden United Kingdom	1991–1998	CD: 75; 54% UC: 177; 64%	CD: 9% obese UC: 12% obese	5	No association between obesity and risk of developing CD (obese vs. normal BMIL OR 0.85, 95% CL 0.29–2.45) or UC (obese vs. normal BMI: OR 1.15, 95% CL 0.62–2.12)
He et al. (2023) [24]	Prospective cohort study	United Kingdom	2006–2010	CD: 915, 56.4% UC: 2039, 47.1%	N/A	12.5	High BMI (HR: 1.18, 95% CL 1.04–1.32) and waist circumference (HR: 1.30, 95% CL 1.14–1.49) showed positive association with increased risk of IBD after adjustment for important confounders
Harpsøe et al. (2015) [25]	Population-based cohort study	Denmark		CD: 138, 100% UC: 394, 100%	8.1% obese	11	Women who were obese (based on pre-pregnancy body weight) had a 1.9-fold increased risk of developing CD (OR 1.88, 95% CL 1.02–3.47) No association between pre-pregnancy obesity and risk of developing UC (OR 0.77, 95% CL 0.48–1.25)
Pringle et al. (2015) [26]	Cross-sectional study	USA	2004	CD: 846, 55%	16% obese	12	Obese patients had lower prevalence of penetrating disease compared to overweight adults with BMI < 25 kg/m^2^ (OR 0.56, 95% CL 0.31–0.99) Genetic predisposition did not modify the effect of obesity on CD-related complications
Hass et al. (2006) [27]	Retrospective cohort study	USA	1997–2002	CD: 138, 59%	32% obese	13	No difference in the number of CD-related surgeries or escalation of medical therapy in overweight and obese patients, compared to patients with BMI < 25 kg/m^2^ No difference in the time to first surgery between patients who were overweight compared with patients with BMI < 25 kg/m^2^.
Seminerio et al. (2015) [19]	Retrospective cohort study	USA	2009–2011	CD: 860, 51% UC: 634, 51%	31.5% obese	N/A	Patients with BMI ≥ 35 kg/m^2^ have reduced IBD-related quality of life and higher frequency of elevated CRP compared to patients with normal BMI. No difference in risk of IBD-related surgery, hospitalisation, emergency department presentation in patients who were obese vs. normal BMI
Flores et al. (2015) [18]	Retrospective cohort study	USA	2000–2012	CD: 297, 31% UC: 284, 24%	CD: 30.3% obese UC: 35.2% obese	5.2	No differences in disease extent in UC and disease phenotype in CD amongst obese patients Lower rates of composite endpoint of CD and UC related surgery or hospitalisation and/or initiation of anti-TNF therapy in obese and overweight patients compared to patients with normal BMI or underweight

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
