# Peer review of "The Impact of Obesity on Inflammatory Bowel Disease"

_biomedicines, 2023, doi:10.3390/biomedicines11123256_

Round 1

Reviewer 1 Report

Comments and Suggestions for Authors

The manuscript by Kaazan et al discussed the association between obesity and IBD. The authors provided a comprehensive description of the pathology of obesity and its potential impacts on IBD development and outcomes. Additionally, the table and figure offered a good summary of the content. However, several points need to be improved as detailed below:

1.       There are several typos and grammar mistakes that need to be corrected.

2.       The discussion on obesity and disease management (line 220-223) is superficial and lacks critical information. It would be beneficial to expand on this topic.

Comments on the Quality of English Language

There are several typos and grammar mistakes that need to be corrected.

Author Response

Thank you for your time and review

   1.    There are several typos and grammar mistakes that need to be corrected: Reviewed and fixed as possible 

  1. The discussion on obesity and disease management (line 220-223) is superficial and lacks critical information. It would be beneficial to expand on this topic: The discussion was expanded with more references included - both in the main short  paragraph "obesity and disease management" and in the TNF-a section 

All highlighted in the main manuscript

Kind regards,

Reviewer 2 Report

Comments and Suggestions for Authors

"The Impact of Obesity on Inflammatory Bowel Disease" is an interesting review with few striking evidences.

 1, In this review, author aimed to highlight the epidemiology of obesity in IBD and to describe its influence on disease course and outcomes. A role for obesity in the pathogenesis of IBD was obscure and results from available evidences are conflicting (table 1). It does however appear to negatively influence disease course whilst impacting on the medical and surgical therapies.

2, The pro-inflammatory profile of the visceral adipose tissue might play a role in the pathogenesis and course of Crohn’s Disease (CD).

3, Surgical therapies are discussed but points of discussions are obscure concerning how to tret obesity or adipose tissue mass.

Anti-obesity therapy is not widely used in IBD but was suggested as an adjunctive therapy in those patients and there might be increasing evidences in the near future.

Comments on the Quality of English Language

There are some typographical errors.

Author Response

Thank you for your time and your review.

 - Surgical therapies are discussed but points of discussions are obscure concerning how to treat obesity or adipose tissue mass - attached in the surgical treatment section and highlighted in red.

-Typo errors addressed